# Learning Interpretable Characteristic Kernels via Decision Forests

## Abstract

Decision forests are popular tools for classification and regression. These forests naturally generate proximity matrices that measure the frequency of observations appearing in the same leaf node. While other kernels are known to have strong theoretical properties such as being characteristic, there is no similar result available for decision forest-based kernels. In addition, existing approaches to independence and k-sample testing may require unfeasibly large sample sizes and are not interpretable. In this manuscript, we prove that the decision forest induced proximity is a characteristic kernel, enabling consistent independence and k-sample testing via decision forests. We leverage this to introduce kernel mean embedding random forest (KMERF), which is a valid and consistent method for independence and k-sample testing. Our extensive simulations demonstrate that KMERF outperforms other tests across a variety of independence and two-sample testing scenarios. Additionally, the test is interpretable, and its key features are readily discernible. This work therefore demonstrates the existence of a test that is both more powerful and more interpretable than existing methods, flying in the face of conventional wisdom of the trade-off between the two.

## 1 Introduction

Decision forests are ensemble method popularized by Breiman [3]. It is highly effective in classification and regression tasks, particularly in high-dimensional settings [5, 6, 37]. This is achieved by randomly partitioning the feature set and using subsampling techniques to construct multiple decision trees from the training data. To measure the similarity between two observations, a proximity matrix can be constructed, defined as the percentage of trees in which both observations lie in the same leaf node [4]. This proximity matrix serves as an induced kernel or similarity matrix for the decision forest. In general, any random partition algorithm may produce such a kernel matrix.

As the complexity of datasets grow, it becomes increasingly necessary to develop methods that can efficiently perform independence and k-sample testing. We also desire methods that are interpretable, lending insight into how and why statistically significant results were determined. Parametric methods are often highly interpretable, such as Pearson's correlation and its rank variants [22, 33, 16]. These methods are still popular to detect linear and monotonic relationships in univariate settings, but they are not consistent for detecting more complicated nonlinear relationships. Nonparametric methods can be very powerful. The more recent distance correlation (Dcorr) [36, 35] and the kernel correlation (HSIC) [10, 11] are consistent for testing independence against any distribution of finite second moments for any finite dimensionality; moreover, the energy-based statistics (such as Dcorr) and kernel-based statistics (such as HSIC) are known to be exactly equivalent for all finite samples [21, 30]. The theory supporting universal consistency of these methods (which we refer to as kernel methods hereafter, without loss of generality) depends on those kernels being characteristic

kernel [28, 18, 19, 30]. Unfortunately, the above tests do not attempt to further characterize the dependency structure. To the best of our knowledge, very few tests exist[38, 15].

In addition, although these methods all have asymptotic guarantees, for finite samples, performance can be impaired by poorly choosing a particular characteristic kernel. Choosing an appropriate kernel that properly summarize geometries within the data is often times non-obvious [27]. High-dimensional data is particularly vexing [25, 38], and a number of extensions have been proposed to achieve better power such as adaptive metric kernel choice [12], low-dimensional projections [14], and marginal correlations [29].

In this paper, we leverage the popular random forest method [3] and a recent chi-square test [32] for a more powerful and interpretable method for hypothesis testing. We prove that the random forest induced kernel is a characteristic kernel, and the resulting kernel mean embedding random forest (KMERF) is a valid and consistent method for independence and k-sample testing. We then demonstrate its empirical advantage over existing tools for high-dimensional testing in a variety of dependence settings, suggesting that it will often be more powerful than existing approaches in real data. As random forest can directly estimate feature importances [3], the outputs of KMERF are also interpretable, KMERF therefore flies in the face of conventional wisdom that one must choose between power and interpretability: KMERF is both empirically more powerful and more interpretable than existing approaches.

## 2 Preliminaries

### 2.1 Hypothesis Testing

The testing independence hypothesis is formulated as follows: suppose $x_i \in \mathbb{R}^p$ and $y_i \in \mathbb{R}^q$, and $n$ samples of $(x_i, y_i) \overset{iid}{\sim} F_{XY}$, i.e., $x_i$ and $y_i$ are realizations of random variables $X$ and $Y$. The hypothesis for testing independence is

$$H_0 : F_{XY} = F_X F_Y,$$
$$H_A : F_{XY} \neq F_X F_Y.$$

Given any kernel function $k(\cdot, \cdot)$, we can formulate the kernel induced correlation measure as $c_k^n(\mathbf{x}, \mathbf{y})$ using the sample kernel matrices [10, 30], where $\mathbf{x} = \{x_i\}$ and $\mathbf{y} = \{y_i\}$. When the kernel function $k(\cdot, \cdot)$ is characteristic, it has been shown that $c_k^n(\mathbf{x}, \mathbf{y}) \to 0$ if and only if $\mathbf{x}$ and $\mathbf{y}$ are independent [10].

The k-sample hypothesis is formulated as follows: let $u_i^j \in \mathbb{R}^p$ be the realization of random variable $U_j$ for $j = 1, \ldots, l$ and $i = 1, \ldots, n_j$. Suppose the $l$ datasets that are sampled i.i.d. from $F_1, \ldots, F_l$ and independently from one another. Then,

$$H_0 : F_1 = F_2 = \cdots = F_l,$$
$$H_A : \exists\, j \neq j' \text{ s.t. } F_j \neq F_{j'}.$$

By concatenating the $l$ datasets and introducing an auxiliary random variable, the kernel correlation measure can be used for k-sample testing [21].

### 2.2 Characteristic Kernel

**Definition 1.** *Let $\mathcal{X}$ be a separable metric space, such as $\mathbb{R}^p$. A kernel function $k(\cdot, \cdot) : \mathcal{X} \times \mathcal{X} \to \mathbb{R}$ measures the similarity between two observations in $\mathcal{X}$, and an $n \times n$ kernel matrix for $\{x_i \in \mathcal{X}, i = 1, \ldots, n\}$ is defined by $\mathbf{K}(i, j) = k(x_i, x_j)$.*

- *A kernel $k(\cdot, \cdot) : \mathcal{X} \times \mathcal{X} \to \mathbb{R}$ is positive definite if, for any $n \geq 2$, $x_1, \ldots, x_n \in \mathcal{X}$ and $a_1, \ldots, a_n \in \mathbb{R}$, it satisfies*

$$\sum_{i,j=1}^{n} a_i a_j k(x_i, x_j) \geq 0.$$

- *A characteristic kernel is a positive definite kernel that has the following property: for any two random variables $X_1$ and $X_2$ with distributions $F_{X_1}$ and $F_{X_2}$,*

$$E[k(\cdot, X_1)] = E[k(\cdot, X_2)] \text{ if and only if } F_{X_1} = F_{X_2}. \tag{1}$$

## 3 KMERF

The proposed approach for hypothesis testing, KMERF, involves the following steps:

1. Run random forest with $m$ trees, with independent bootstrap samples of size $n_b \leq n$ used to construct each tree. The tree structures (partitions) within the forest $\mathbf{P}$ are denoted as $\phi_w \in \mathbf{P}$, where $w \in 1, \ldots, m$ and $\phi_w(x_i)$ represents the partition assigned to $x_i$.

2. Calculate the proximity kernel by

$$\mathbf{K}_{ij}^{\mathbf{x}} = \frac{1}{m} \sum_{w=1}^{m} [\mathbf{I}(\phi_w(x_i) = \phi_w(x_j))],$$

where $\mathbf{I}(\cdot)$ is the indicator function that checks whether the two observations lie in the same partition in each tree.

3. Compute the unbiased kernel transformation [34, 32] on $\mathbf{K}^{\mathbf{x}}$. Namely, let

$$\mathbf{L}_{ij}^{\mathbf{x}} = \begin{cases} \mathbf{K}_{ij}^{\mathbf{x}} - \frac{1}{n-2} \sum_{t=1}^{n} \mathbf{K}_{it}^{\mathbf{x}} - \frac{1}{n-2} \sum_{s=1}^{n} \mathbf{K}_{sj}^{\mathbf{x}} + \frac{1}{(n-1)(n-2)} \sum_{s,t=1}^{n} \mathbf{K}_{st}^{\mathbf{x}} & i \neq j \\ 0 & i = j \end{cases}$$

4. Let $\mathbf{K}^{\mathbf{y}}$ be the Euclidean distance induced kernel by Shen and Vogelstein [30], or the proximity kernel in the case that dimensions of $\mathbf{x}$ and $\mathbf{y}$ is the same, that is $p = q$, and compute $\mathbf{L}^{\mathbf{y}}$ using the same unbiased transformation. Then the KMERF statistic for the induced kernel $k$ is,

$$c_k^n(\mathbf{x}, \mathbf{y}) = \frac{1}{n(n-3)} \text{trace}(\mathbf{L}^{\mathbf{x}} \mathbf{L}^{\mathbf{y}}).$$

5. Compute the p-value via the following chi-square test [32]:

$$p = 1 - F_{\chi_1^2 - 1} \left( n \cdot \frac{c_k^n(\mathbf{x}, \mathbf{y})}{\sqrt{c_k^n(\mathbf{x}, \mathbf{x}) \cdot c_k^n(\mathbf{y}, \mathbf{y})}} \right),$$

where $\chi_1^2$ is the chi-square distribution of degree 1. Reject the independence hypothesis if the p-value is less than a specified typer 1 error level, say 0.05.

In the numerical implementation, the standard supervised random forest is used with $m = 500$ (which is also applicable to the unsupervised version or other random forest variants [2, 1, 37]). In the second step, we simply compute the proximity kernel defined by the random forest induced kernel. In the third step, we normalize the proximity kernel to ensure it obtains a consistent dependence measure; this is the KMERF test statistic. We found that utilizing the multiscale version of the kernel correlation [38, 31], which is equivalent for linear relationships while being better for nonlinear relationships, produced similar results to using distance correlation, but substantially increased runtimes.

Note that one could also compute a p-value for KMERF via the permutation test, which is a standard procedure for testing independence [9]. Specifically, first compute a kernel on the observed $\{x_i\}$ and $\{y_i\}$. Then randomly permute the index of $\{y_i\}$, repeat the kernel generation process for $\{y_i\}$ for each permutation. This process involves training a new random forest for each permutation. Finally, compute the test statistic for each of the permutations, and the p-value equals the percentage the permuted statistics that are larger than the observed statistic. However, the permutation test is very slow for large sample size and almost always yields similar results as the chi-square test.

## 4 Theoretical Properties

Here, we show that the random forest kernel characteristic, and the induced test statistic used in KMERF allows for valid and universally consistent independence and k-sample testing. All proofs are in appendix.

For a kernel to be characteristic, it first needs to be positive definite, which is indeed the case for the forest-induced kernel:

**Theorem 1.** *The random forest induced kernel $\mathbf{K}^{\mathbf{x}}$ is always positive definite.*

This theorem holds because the forest-induced kernel is a summation of a permuted block diagonal matrix, with each matrix coming from individual tree, that is positive definite [7]; and a summation of positive definite matrices is still positive definite.

Next, we show the kernel is characteristic when the tree partition area converges to zero. A similar property is also used for proving classification consistency for k-nearest-neighbors [8], and we shall denote $N(\phi_w)$ as the maximum area of each part.

**Theorem 2.** *Suppose as $n, m \to \infty$, $N(\phi_w) \to 0$ for each tree $\phi_w \in \mathbf{P}$ and each observation $x_i$. Then the random forest induced kernel $\mathbf{K}^{\mathbf{x}}$ is asymptotically characteristic.*

Intuitively, for sufficiently many trees and sufficiently small leaf region, observations generated by two different distributions cannot always be in the same leaf region.

This leads to the validity and consistency result of KMERF:

**Corollary 2.1.** *KMERF satisfies*

$$\lim_{n \to \infty} c_k^n(\mathbf{x}, \mathbf{y}) = c \geq 0,$$

*with equality to $0$ if and only if $F_{XY} = F_X F_Y$. Moreover, for sufficiently large $n$ and sufficiently small type 1 error level $\alpha$, this method is valid and consistent for independence and k-sample testing.*

By Gretton et al. [10], any characteristic-kernel based dependence measure converges to $0$ if and only if $X$ and $Y$ are independent. Moreover, Shen et al. [32] showed that the chi-square distribution $\chi_1^2 - 1$ approximates and upper-tail dominates the true null distribution of any unbiased kernel when using distance correlation, making it a valid and consistent test.

## 5 Simulations

In this section we exhibit the consistency and validity of KMERF, and compare its testing power with other competitors in a comprehensive simulation set-up. We utilize the `hyppo` package in Python [20], which uses `scikit-learn` [23] random forest with $500$ trees and otherwise default hyper-parameters, and calculate the proximity matrix from this. The KMERF statistic and p-value then computed via the process in Section 3. The mathematical details for each simulation type is in the Appendix C.

### 5.1 Testing Independence

In this section we compare KMERF to Multiscale Graph Correlation (MGC), Distance Correlation (Dcorr), Hilbert-Schmidt Independence Criterion (Hsic), and Heller-Heller-Gorfine (HHG) method, Canonical Correlation Analysis (CCA), and the RV coefficient. The HHG method has been shown to work extremely well against nonlinear dependencies [13]. The MGC method has been shown to work well against linear, nonlinear, and high-dimensional dependencies [31]. The CCA and RV coefficients are popular multivariant extensions of Pearson correlation. For each method, we use the corresponding implementation in `hyppo` with default settings.

We take 20 high-dimensional simulation settings [38], consisting of various linear, monotone, and strongly nonlinear dependencies with $p$ increasing, $q = 1$, and $n = 100$. To estimate the testing power in each setting, we generate dependent $(x_i, y_i)$ for $i = 1, \ldots, n$, compute the test statistic for each method, repeat for $r = 10000$ times. Via the empirical alternative and null distribution of the test statistic, we estimate the testing power of each method at type 1 error level of $\alpha = 0.05$. The power result is shown in Figure 1 shows that KMERF achieves superior performance for most simulation modalities, except a few like circle and ellipse.

### 5.2 Two Sample Testing

Here, we compare the performance in the two-sample testing regime. It has been shown that all independence measures can be used for two-sample testing [21, 30], allowing all previous independence testing methods to be compared here as well. Once again, we investigate statistical

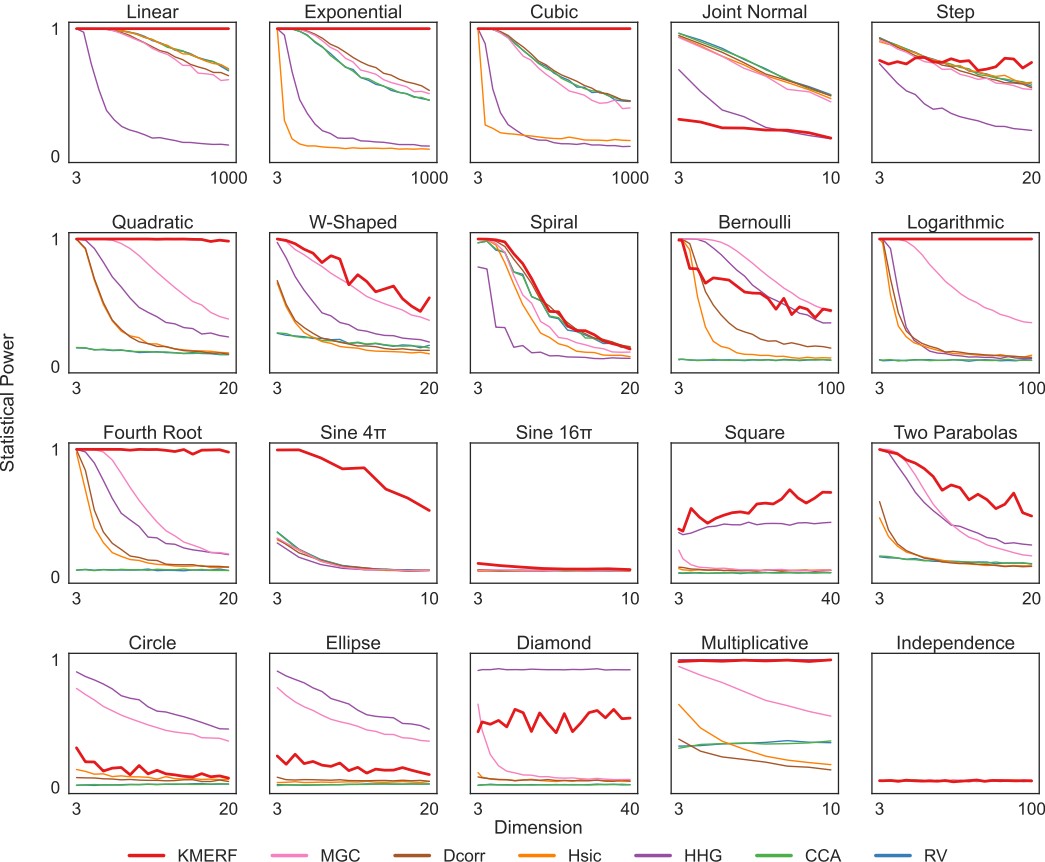

Figure 1: Multivariate independence testing power for 20 different settings with increasing $p$, fixed $q = 1$, and $n = 100$. For the majority of the simulations and simulation dimensions, KMERF performs as well as, or better than, existing multivariate independence tests in high-dimensional dependence testing.

power differences with 20 simulation settings consisting of various linear and nonlinear, monotonic and nonmonotonic functions with dimension increasing from $p = 3, \dots, 10$, $q = 1$, and $n = 100$. We then apply a random rotation to this generated simulation and generate the second independent sample (via a rigid transformation).

Figure 2 shows that, once again, for the majority of simulations settings, KMERF performs at or better than other tests in nearly all simulations and simulation dimensions. For certain simulation settings, especially the exponential, cubic, and fourth root, KMERF vastly outperforms other metrics as dimensions increases.

## 5.3 Interpretability

Not only does KMERF typically offer empirically better statistical power compared to alternatives, it also offers insights into which features are the most important within the data set. Figure 3 shows normalized 95% confidence intervals of relative feature importances for each simulation, where the black line shows the mean and the light grey line shows the 95% confidence interval. Mean and individual tree feature importances were normalized using min-max feature scaling. The simulations were modified such that the weighting of each feature decreased as feature importance increased, with the expectation that the algorithm would detect a decrease in feature importance as dimension increased. With these simulations, we are able to determine that exact feature importance trend,

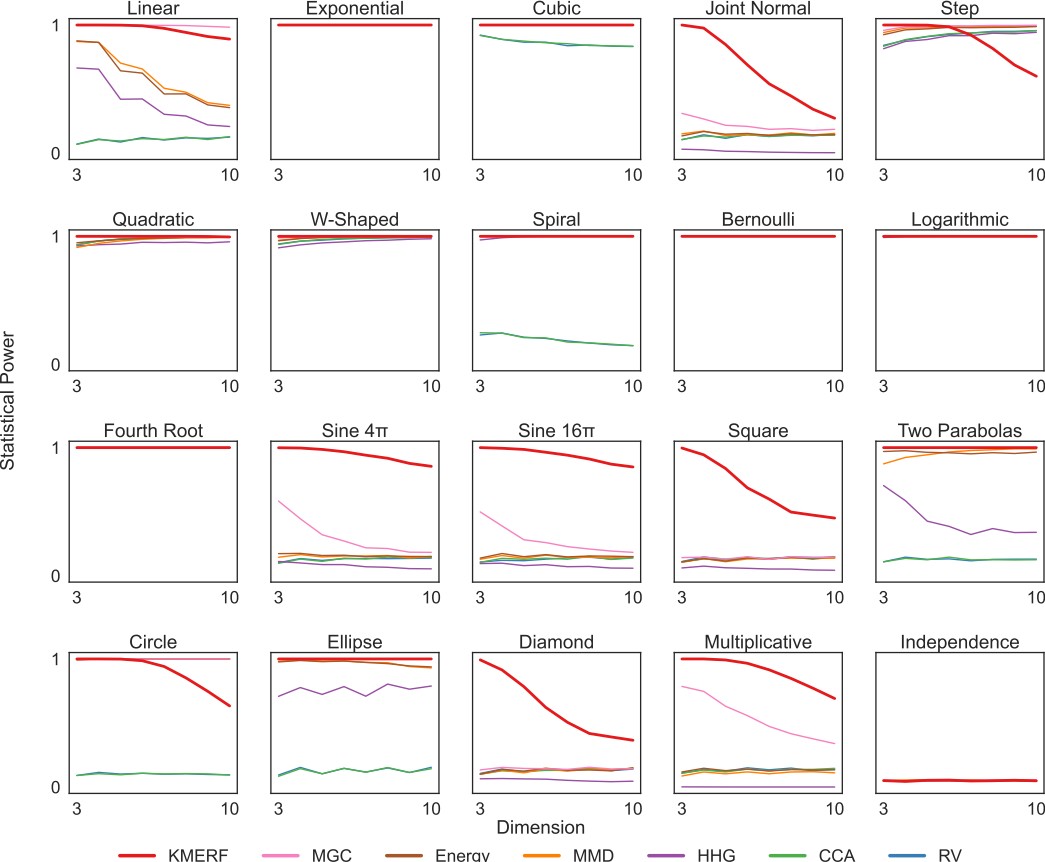

Figure 2: Multivariate two-sample testing power for 20 different settings with increasing $p$, fixed $q = 1$, and $n = 100$. For nearly all simulations and simulation dimensions, KMERF performs as well as, or better than, existing multivariate two-sample tests in high-dimensional dependence testing.

except for a few of the more complex simulations. The process we used to generate this figure can be trivially extended to a two-sample or k-sample case.

## 6   Real Data

We then applied KMERF to a date set consisting of proteolytic peptides derived from the blood samples of 95 individuals harboring pancreatic (n=10), ovarian (n=24), colorectal cancer (n=28), and healthy controls (n=33) [38]. The processed data included 318 peptides derived from 121 proteins (see Appendix D for full details). Figure 4 shows the p-values for KMERF between pancreatatic and healthy subjects compared to the p-values for KMERF between pancreatic cancer and all other subjects. The test identifies neurogranin as a potentially valuable marker for pancreatic cancer, which the literature also corroborates [41, 40]. Meanwhile, while some of the other tests identified this biomarker, they identified others that are upregulated in other types of cancers as well (false positives). We also show in the figure that the biomarker chosen be KMERF provides better true positive detection when compared to the other tests (there is no ground truth in this case, so a leave-one-out k-nearest-neighbor classification approach was used instead).

Feature Importances

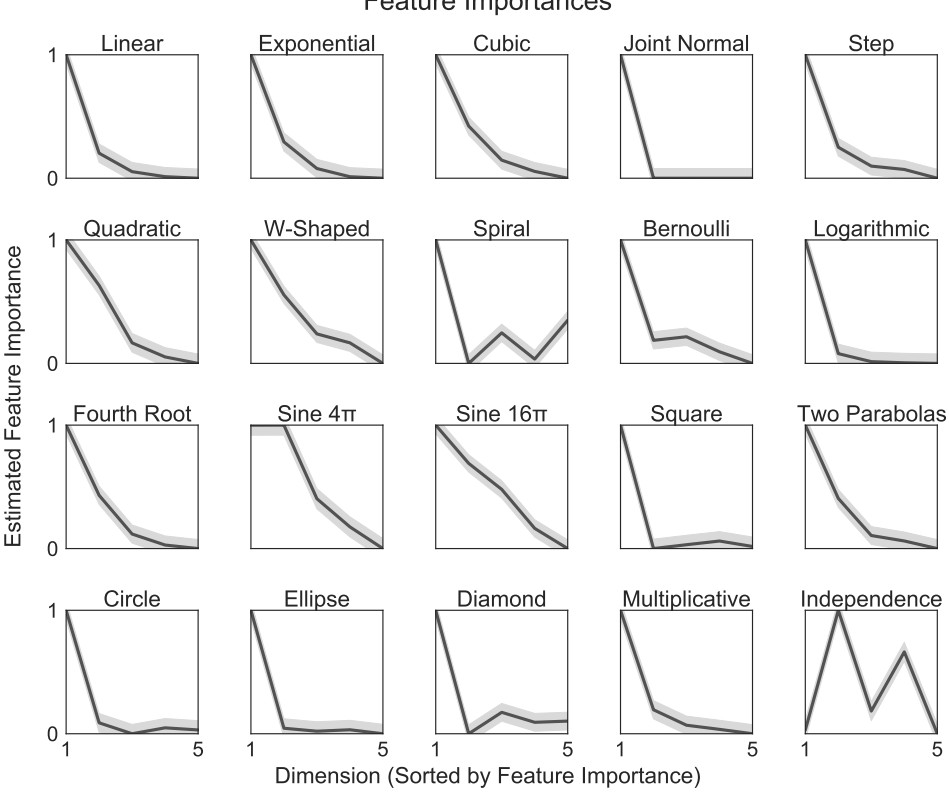

Figure 3: Normalized mean (black) and 95% confidence intervals (light grey) using min-max normalization for relative feature importances derived from random forest over five dimensions for each simulation tested for 100 samples. The features were sorted from most to least informative for all simulations except for the Independence simulation). As expected, estimated feature importance decreases as dimension increases. A feature of KMERF is insights into interpretability, and we show here which dimensions of our simulations influence the outcome of independence test the most.

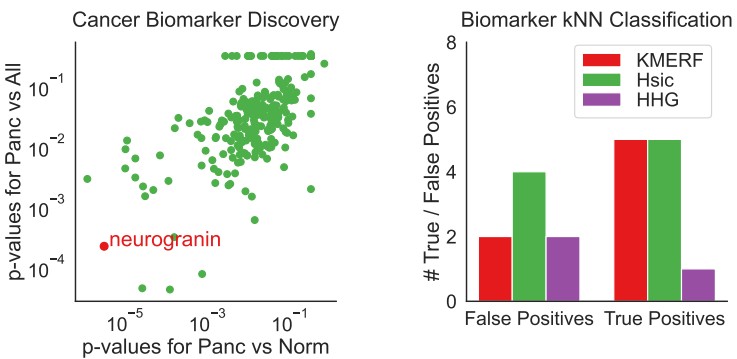

Figure 4: (A) For each peptide, the p-values for testing dependence between pancreatic and healthy subjects by KMERF is compared to the p-value for testing dependence between pancreatic and all other subjects. At the critical level 0.05, KMERF identifies a unique protein. (B) The true and false positive counts using a k-nearest neighbor (choosing the best $k \in [1, 10]$) leave-one-out classification using only the significant peptides identified by each method. The peptide identified by KMERF achieves the best true and false positive rates.

## 7 Discussion

KMERF is, to the best of our knowledge, one of the first learned kernel that is proven to be characteristic. The empirical experiments presented here illustrate the potential advantages of learning kernels, specifically for independence and k-sample testing.

In fact, multiscale graph correlation [38, 31] can be thought of, in a sense, as kernel learning: given $n$ samples, and a pair of kernel or distance functions, it chooses one of the approximately $n^2$ sparsified kernels, by excluding all but the nearest neighbors for each data point [38, 31]. Because random forest can be thought of as a nearest neighbor algorithm [17], in a sense, the forest induced kernel is a natural extension of Vogelstein et al. [38], which leads to far more data-adaptive estimates of the nearest neighbors using supervised information. Moreover, proving that the random-forest induced kernel is characteristic is a first step towards building lifelong learning kernel machines with strong theoretical guarantees [24, 39].

As the choice of kernel is crucial for empirical performance, this manuscript offers a new kernel construction that is not only universally consistent for testing independence, but also exhibits strong empirical advantages, especially for high-dimensional testing. What is unique to this choice of kernel is the robustness and interpretability. It will be worthwhile to further understand the underlying theoretical mechanism of the induced characteristic kernel, as well as evaluating the performance of these forest induced kernels on other learning problems, including classification, regression, clustering, and embedding [26].

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
