# OpenReview forum: "Learning Interpretable Characteristic Kernels via Decision Forests"
_NeurIPS.cc/2023/Conference — Submitted to NeurIPS 2023_

### Official Review · Reviewer_11WJ · 2023-07-02

**Soundness:** 2 fair
**Presentation:** 2 fair
**Contribution:** 2 fair
**Rating:** 5
**Confidence:** 3

**Summary:**

The authors show the random forest induced kernel is characteristic, and they empirically study the validity of the kernel for independence and k-sample testing.

**Strengths:**

The authors show the connection between the characteristic property of the kernel and random forest. The topic is interesting and important.

**Weaknesses:**

The presentation should be improved. Since detailed explanations about random forest are missing, how we can connect the kernel methods to random forest is not clear. I think that is the most important background of this paper, so it should be explained more. In addition, the novelty of the paper is not clear to me since the theoretical results in this paper seem to depend strongly on the random forest setting, but the setting is not clearly explained.

**Questions:**

- In Theorem 2, does $\phi_w$ become injective by the construction of the kernel through random forest? I think specific properties coming from the random forest setting should be explained clearly in the main text.

Minor comments:
- In the supplementary material, there are so many CSV files, but we only need codes and a PDF file for the appendix of the paper.

**Limitations:**

As I also mentioned in the weakness part, since the setting and the background are not sufficiently explained, the novelty and the nontrivial part of this paper are not clear.

---

> ### Author Rebuttal · Authors · 2023-08-09
>
> Thank you very much for the thorough review!
>
> **Weakness and Limitation:**
>
> We will incorporate more background information on random forest kernels and existing works related to hypothesis testing. Moreover, we will include an updated conclusion to emphasize the main contribution of the paper, i.e., the direct utilization of this proximity matrix as a valid and consistent kernel choice for hypothesis testing – a approach not explored in existing literature.
>
>
> Here is the proposed changes for the first paragraph in the introduction section (Line 23-24) to provide a better review of random forest works:
>
> > …This proximity matrix functions as an induced kernel or similarity matrix for the decision forest. The connection between random forest and the induced kernel is well-established, with research demonstrating that random forest can essentially be perceived as a kernel method, generating kernels that surpass conventional ones [1,2, 3]. Generally, any random partition algorithm has the potential to generate a proximity matrix and be conceptualized as a kernel.
>
> Then by the end of the third paragraph (Line 44) we will add the following on existing works regarding random forest for testing.
>
> > ... There are existing studies that utilize random forests for hypothesis testing, such as F-test and feature screening [4], as well as two-sample testing [5]. However, these studies utilize other random forest outputs rather than the proximity matrix. To the best of our knowledge, there has been no research that explores the characteristic properties of the induced kernel, nor any work that directly employs the induced random forest kernel for tasks related to independence or two-sample testing.
>
> Then in the discussion section at the end of the paper (Lines 198-199), we will emphasize the major contribution, and the potential for other random forest algorithms:
>
> > ... The primary contribution of the paper lies in directly leveraging the induced kernel from random forest for achieving consistent and valid hypothesis testing for independence and two-sample. We explored the theoretical properties of this approach while showcasing its numerical benefits. With the aid of the recent chi-square test procedure for kernel-based tests, the complexity of our proposed method is
>  to compute the test statistic and p-value. Additionally, owing to the data-adaptive nature of the random forest algorithm, the resulting kernel appears to effectively adjust to the inherent data structure, thereby enhancing testing power. Other kernels, whether derived from variations of the standard random forest [1,2] or from the proximity matrix of alternative random partition or forest algorithms [6, 7], may also be used in the framework.
>
> **Additional references:**
>
> 1. Alex Davies, Zoubin Ghahramani, “The Random Forest Kernel and creating other kernels for big data from random partitions”, arXiv:1402.4293.
> 2. Erwan Scornet, “Random forests and kernel methods”, arXiv:1502.03836.
> 3. Gerard Biau and Erwan Scornet, "A random forest guided tour”, TEST, 2016.
> 4. Tim Coleman, Wei Peng, Lucas Mentch, “Scalable and Efficient Hypothesis Testing with Random Forests ”, Journal of Machine Learning Research.
> 5. Simon Hediger, Loris Michel, Jeffrey Näf, “On the use of random forest for two-sample testing”, Computational Statistics and Data Analysis, 2022.
> 6. Tyler Tomita et al., “Sparse Projection Oblique Randomer Forests”, Journal of Machine Learning Research, 2020.
> 7. Pierre Geurts, Damien Ernst, Louis Wehenkel, “Extremely randomized trees”, Machine Learning, 2006.
>
>
> **Question 1:**
>
> Regarding the theorem question, yes, $\phi_w$ is injective due to construction of the random forest — as the area of each leaf region goes to zero, two different observations must lie on two different leaf regions as $n$ increases, effectively making them injective for continuous random variables.
>
> We will include the following brief explanation in the theorem section on why the leaf region goes to zero, and make it clear that the assumptions require properties from random forest construction:
>
> > Note that a crucial assumption in this context is that the area of each leaf region approaches zero. This particular asymptotic behavior is generally met by most continuous and smooth random variables when employing standard random forest. For instance, consider a scenario where the sample data has a continuous distribution within the support. In the default configuration of a random forest, each leaf node accommodates a relatively small number of observations, typically 1 in regression or 5 in classification. Consequently, as the sample size increases to infinite, the area of each leaf node converges to zero.
>
> **Question 2:**
>
> Considering the extended runtime required for generating power curve plots, involving numerous settings, multiple sample size intervals, and numerous replicates, we decided to provide the power data in CSV files for the initial submission. We hoped this may facilitate potential figure reproduction and result replication for running the code. Subsequently, we will remove these CSV files, retaining only the code. Note that both the code and CSV files will be accessible on GitHub for public reference later.

---

> > ### Comment · Reviewer_11WJ · 2023-08-14
> >
> > Thank you for your response. I'm still concerned about readability. I wonder formal explanations (not just by giving references, but by giving mathematical definitions and formulae) of the setting of random forest and relation between random forest and kernel methods are missing.

---

> > > ### Author Response · Authors · 2023-08-18
> > >
> > > Yes, we agree that formal definitions of random forest and the kernel will enhance readability. We feel that this is very important, and so propose adding a new section 2.3 in the Preliminaries:
> > >
> > > ### Random Forest and the Proximity Kernel
> > >
> > > Given a dataset of $(x_i, y_i)$ for $i = 1, \ldots, n$, we employed the standard Classification and Regression Trees (CART) algorithm (1, 2) to build each tree. In the context of an independence test, where $y_i$ is continuous, the construction of the tree involves identifying the optimal dimension $j$ and the corresponding optimal split point $s$ that satisfies the minimization problem:
> > > $$
> > > \min_{j, s} \left[ \min_{c_a} \sum_{x \in R_a(j,s)} (y_i - c_a)^2 + \min_{c_b} \sum_{x \in R_b(j,s)} (y_i - c_b)^2 \right].
> > > $$
> > > Here, $c_a$ and $c_b$ are the sample means within the regions $R_a$ and $R_b$ respectively. These regions are essentially half-planes within the current region. For instance, for the initial split, $R_a = \\{x\_i|x\_i^{j} \leq s\\}$ and $R_b = \\{x_i|x_i^{j}>s\\}$.
> > >
> > > The criterion above is essentially the mean squared error within each region for regression. When $y_i$ is categorical, such as in the case of two-sample testing, the same algorithm is used, but the minimization is performed over the Gini index to gauge impurity of the region.
> > >
> > > In essence, the CART algorithm identifies the optimal pair $(j,s)$ to iteratively build a tree structure on the sample data, aiming to decrease an objective function within each region, until a predefined stopping criterion is met. By default, the stopping criterion is five observations within each region for regression trees and, in classification, we typically stop when each region is 'pure' (only has samples from a single class). Upon reaching the stopping criterion, the resulting regions within the tree are commonly referred to as leaf nodes.
> > >
> > > Random forests are  ensembles of CARTs (3,4). Each tree is constructed by resampling the training data with replacement. In our experiments, we used 500 trees. The time complexity of random forests is $\mathcal{O}(mpn \log(n))$, where $m$ is the number of trees, $p$ is the dimensionality of the data, and $n$ is the sample size.
> > >
> > > Finally, the standard proximity kernel is computed as follows:
> > >
> > > **Definition 2.** *Given a forest of $m$ trees, let $\phi_w \in \mathbf{P}, w \in 1, \ldots, m$ denote a single decision tree. The proximity kernel matrix is computed by
> > > $$\mathbf{K}^{\mathbf{x}}\_{ij} = \frac{1}{m}\sum\limits_{w = 1}^{m}[\mathbb{1}(\phi_w(x_i) = \phi_w(x_j))],$$
> > > where $\mathbb{1}$ is the indicator function that observations $x_i, x_j \in \mathbf{x}$ are in the same leaf node in each tree.*
> > >
> > > The proximity kernel lies in $[0,1]$. As a simple example, if $x_i$ and $x_j$ always lie in the same leaf node in every tree, then $\mathbf{K}^{\mathbf{x}}\_{ij}=1$. If out of  $100$ trees, there are only $10$ trees where $x_i$ and $x_j$ belong to the same leaf node, $\mathbf{K}^{\mathbf{x}}\_{ij}=0.1$.
> > >
> > > **References:**
> > > 1. Leo Breiman, "Classification and regression trees", Routledge, 1984.
> > > 2. Trevor Hastie, Robert Tibshirani, Jerome Friedman, "Elements of Statistical Learning", Springer, 2001.
> > > 3. Leo Breiman, "Random Forests", Machine Learning, 2001.
> > > 4. Leo Breiman, "Bagging predictors", Machine Learning, 1996.

---

> > > > ### Comment · Reviewer_11WJ · 2023-08-19
> > > >
> > > > Thank you for your response. Based on the response, I raised my score.

---

### Official Review · Reviewer_h7qn · 2023-07-04

**Soundness:** 3 good
**Presentation:** 1 poor
**Contribution:** 2 fair
**Rating:** 4
**Confidence:** 1

**Summary:**

The paper show how decision forest can be used to induce a kernel for k-sample testing.

**Strengths:**

Better-than-SOTA results.

**Weaknesses:**

The paper is a extremely hard to follow, particularly Section 3, which I was unable to follow without going deep into the referenced papers. The same goes for related works, which are simply mentioned en-passant in Section 5.1. A reader not directly familiar with all of them is left wondering in what they differ, and why.
Additional comments:
- L18: "are an ensemble".
- L18: "They are highly effective".
- Given that we're dealing with matrices, I'd suggest using \mathds{1} to indicate indicator functions, rather than I, which is usually used for identity matrices.
- Figure 4 only reports 2 competitors: what about the others?


**Questions:**

Could not formulate questions due to difficulty in understanding the paper.

**Limitations:**

Limitedly comprehensible.

---

> ### Author Rebuttal · Authors · 2023-08-09
>
> Thank you for taking the time to review the paper!
>
> **Additional Comments 1-3:**
>
> Thank you for pointing out the typos, we fixed them and also changed notation for the indicator matrix.
>
> **Additional Comments 4:**
>
> Figure 4 exclusively showcased two competitors, HHG and HSIC, for specific reasons: MGC detected the same peptide as KMERF, and HHG and HSIC identified a slightly distinct set of significant peptides. On the other hand, all other methods (DCor, CCA, RV) failed to identify any significant peptides. To provide more clarity, see updated Figure 3 in the attached PDF. This figure contains a more comprehensive caption, along with the addition of the base case where all peptides (dimensions) are incorporated into the classification task, representing all other methods.
>
> **Weaknesses:**
>
> We sincerely appreciate your feedback regarding Section 3 and Section 5. While the detailed background information on these sections was initially included in the draft, it was regrettably omitted from the initial submission due to space constraints. In our revised draft, we will ensure to reinstate this information within the appendix for comprehensive reference.
>
> For the main method in Section 3, we will provide detailed information regarding each step. Specifically, we will provide the following:
> - in step 1, we will explain the parameters and the default implementation utilized for the standard random forest.
> - in step 3, we will present the motivation and rationale underpinning the use of the unbiased transformation. This transformation serves to achieve $E[corr]=0$ under conditions of independence, which, in turn, facilitates the successful application of the chi-square test.
> - in step 5, an in-depth explanation will be provided as to why the chi-square test is a fast, valid, and consistent alternative to the permutation test. Furthermore, we will explain the standard permutation test for other methods.
>
> Then, for the competitor methods in section 5.1, the updated appendix will include a thorough exploration of each method, elucidating their known properties and providing concise comparative insights. Specifically, we will incorporate the following information:
> - DCor and HSIC: A detailed account of the statistics and testing process for both DCor and HSIC will be included. It will also be highlighted that DCor and HSIC are fundamentally the same methods, with DCor operating on distance matrices and HSIC on kernel matrices.
> - MGC: It is a local and adaptive version of DCor. We will explain its use of local distance computation and its capability to identify optimal local neighborhoods, rendering it adept at identifying nonlinear relationships.
> - HHG: it is a rank-based method, specifically tailored for capturing nonlinear dependencies effectively.
> - CCA and RV: they are multivariate extensions of Pearson correlation, so their strengths lie primarily in linear dependence scenarios.
>
>
> Overall, the updated appendix will provide the general audience with detailed information on all the methods without the need to go through other references.

---

### Official Review · Reviewer_qW4w · 2023-07-04

**Soundness:** 4 excellent
**Presentation:** 4 excellent
**Contribution:** 4 excellent
**Rating:** 6
**Confidence:** 3

**Summary:**

 In this paper, the authors introduced a new method called KMERF, which employs random forest for kernel construction. Through their algorithm, they were able to establish that the kernel they created has certain properties, namely being positive definite and asymptotically characteristic. The authors also demonstrated that KMERF has better statistical power than other independent methods. This makes KMERF one of the pioneering learned kernels that has been proven to be asymptotically characteristic.

**Strengths:**

I find the proposed method to be impressive and well-organized. The paper is also well-written. It's worth noting that the number of learned kernels with characteristic properties is limited, so KMERF could be a valuable contribution to the kernel learning field. While it's not unexpected for a learned kernel to show better statistical power, the decent false discovery rate of KMERF is intriguing.

**Weaknesses:**

1. There is concern about potential inflation from post-selection inference since the kernel being proposed is learned. To address this, I suggest that the authors conduct additional calibration tests of KMER. For instance, they could create various simulation scenarios (similar to those presented in Fig1 and Fig2) involving a mix of causal and null features, and demonstrate the well-controlled FPR/ FDR.

2. Further details are required in Fig1 and Fig2, such as parameter values and exact equations, to fully understand how each simulation setting was carried out.

3. It would be beneficial to have a more comprehensive analysis of the runtime.

**Questions:**

In supplementary proof theorem 1, line 8, it is not clear to me why "Each block matrix is always positive definite".

**Limitations:**

KMERF has been proven to exhibit asymptotic characteristics when the sample size n and the number of trees m approach infinity. However, in practical applications, m is often limited to prevent overfitting and computational issues. Providing insights on KMERF's behavior under finite n and reasonable m would be greatly beneficial.

---

> ### Author Rebuttal · Authors · 2023-08-09
>
> Thank you very much for the valuable comments and questions. Below is the response to all the weaknesses and questions.
>
> **Weakness 1:**
>
> As shown in Figure 1 and Figure 2, the test power equals the type 1 error under independence. This supports the validity of the test, which means the test will not cause an inflation of the false positive rate during post-selection inference. To substantiate this point, we conducted an additional simulation, the results of which are presented in Figure 1 of the attached document and showed no inflation.
>
> In essence, we generated ($x_1$, $x_2$, $x_3$, $x_4$, $x_5$, $x_6$), where each $x_i$ is drawn independently from a $\mathcal{U}(-1, 1)$ distribution. Then we set $Y = x_1 + x_2^2 + x_3 + \epsilon$, where $\epsilon$ is a noise parameter. This establishes a relationship between the first three variables and $Y$, while the last three variables are independent of $Y$. This is repeated for 100 replicates, and in each replicate we generate sample data, then compute the KMERF statistic and p-value between each of $x_1, \ldots, x_6$ and $Y$.  We set the true positive rate as how often the dependent variables are flagged as significant (p-value < 0.05), and set the false positive rate as how often the independent variables are flagged as significant. The computations were performed for each variable, and we report the average true positive among the first three variables, and the average false positive among the last three variables. As expected, the figure shows that the true positive goes to 1 and the false positive stays at 0.05.
>
>
> **Weakness 2:**
>
> For Figure 1 and 2, the exact parameters and equations for the simulations are included in appendix section C. The specific hyperparameters (n,p,q) employed in each test are specified in the figure captions. The code to generate them and replicate all figures will be publicly available on GitHub.
>
>
>
> **Weakness 3:**
>
> A detailed running time analysis is provided as follows, which will be added to the main paper:
>
> > The KMERF method consists of three major components: random forest, distance correlation computation, and hypothesis testing. For the number of trees ($m$), the number of dimensions ($p$), and the number of samples ($n$), the complexity of the random forest is $\mathcal{O}(mpn\log{n})$, the correlation computation complexity is $\mathcal{O}(n^2)$, and the chi-square testing complexity is $\mathcal{O}(1)$. This overall process is fast and scalable.
>
> For reference, In the attached post-selection figure (Figure 1 in appendix), it took about 45 seconds on a Python Jupyter (2019 Macbook Pro with a 2.3 GHz 8-Core Intel Core i9) to compute all six p-values for the sample data at $n= 2000$.
>
> **Question 1:**
>
> Thank you for pointing out this typo, we have amended the manuscript with the change. Indeed, each block matrix should be positive semidefinite — as the diagonal blocks are matrices of ones, eigenvalues of zero exist.
>
> In the context of kernel theory, a kernel is considered positive definite when $\sum_{i,j} a_i a_j k(i,j) >= 0$. Therefore, when the kernel matrix or proximity matrix is positive semidefinite (psd) for any sample data, the kernel function is positive definite (pd). Note that this slight difference on psd / pd is merely a terminology discrepancy between matrix theory and kernel methods.
>
> **Limitation 1:**
>
> Concerning the number of trees, we have included an additional Figure 2 in the attached PDF. This figure examines the effect of varying values of the number of trees $m$ on the testing power for three distinct relationships. As depicted in the attached figure, it is evident that the number of trees has negligible impacts on the testing power of KMERF for these relationships and choices of $m$.

---

> > ### Comment · Reviewer_qW4w · 2023-08-14
> >
> > The authors have addressed the majority of my concerns and questions. It would become a big plus if the authors can provide more theoretical insight into why this post-selection inference procedure won't incur potential inflation.

---

> > > ### Author Response · Authors · 2023-08-18
> > >
> > > Thank you for raising this insightful question! You are right to have concerns about whether learning labels effect validity of the test. To better justify the validity, we will add the following text into the paper:
> > >
> > > When using the permutation test rather than the chi-square test, any post-selection inference is valid. This would involve calculating $c\_k^{n}(\mathbf{x}, \mathbf{y}\_{\pi})$ for $100$ permutations, wherein $\pi$ is a random permutation of the sample indices. The p-value would then be computed as $Prob(c_k^{n}(\mathbf{x}, \mathbf{y}_{\pi}) > c_k^{n}(\mathbf{x}, \mathbf{y}))$. As permutations effectively break the pairwise dependence between each pair of samples, the resulting permuted statistics closely approximate the null distribution of the test statistic. The only assumptions permutation tests make is that the observations are exchangeable under the null hypothesis (1), which is not violated by learning the kernel.
> > >
> > > In the paper we use the chi-square test in step 5, i.e., compute
> > > $$
> > > p=1-F\_{\chi^{2}\_{1}-1} \left(n \cdot \frac{c_k^{n}(\mathbf{x}, \mathbf{y})}{\sqrt{c_k^{n}(\mathbf{x}, \mathbf{x}) \cdot c_k^{n}(\mathbf{y}, \mathbf{y})}}\right),
> > > $$
> > > where $\chi^{2}_{1}$ is the chi-square distribution of degree $1$.
> > > This step is demonstrated to be approximately valid for kernel correlation using an unbiased kernel in (2).
> > >
> > > In particular, it is derived in (3) that under independence between $(X,Y)$, the limiting distribution of an unbiased kernel correlation satisfies
> > > $$
> > > \left(n \cdot \frac{c\_k^{n}(\mathbf{x}, \mathbf{y})}{\sqrt{c\_k^{n}(\mathbf{x}, \mathbf{x}) \cdot c\_k^{n}(\mathbf{y}, \mathbf{y})}}\right) \stackrel{D}{\rightarrow} \sum\limits\_{i,j=1}^{\infty} w_{ij} (\mathcal{N}\_{ij}^{2}-1),
> > > $$
> > > where the weights satisfy $w_{ij} \in [0,1]$ and $\sum\limits_{i,j=1}^{\infty} w_{ij}^{2} = 1$, and $\mathcal{N}\_{ij}$ are independent standard normal distributions. Then it was proved in (2) that the weighted summation of densities is bounded by a chi-square distribution on the upper tail, leading to
> > > $$
> > > \left(n \cdot \frac{c\_k^{n}(\mathbf{x}, \mathbf{y})}{\sqrt{c\_k^{n}(\mathbf{x}, \mathbf{x}) \cdot c\_k^{n}(\mathbf{y}, \mathbf{y})}}\right) \preceq\_{\alpha} \chi^{2}\_{1}-1
> > > $$
> > > for sufficient large $n$ and sufficiently small $\alpha$. Here $\preceq\_{\alpha}$ means upper tail dominance, i.e., $V \preceq\_{\alpha} U$ in upper tail at probability level $\alpha$ if and only if
> > > $$
> > > F\_{V}(x) \geq F\_{U}(x)
> > > $$
> > > for all $x \geq F\_{U}^{-1}(1-\alpha)$. The above dominance is valid for sufficiently large $n$ and adequately small values of $\alpha$ (numerically verified to be approximately $0.08$). This makes the chi-square test a valid test (and in fact slightly conservative) in practical scenarios.
> > > This leads to the following Lemma:
> > >
> > > **Lemma 1.** *The forest-induced kernel satisfies the assumptions of Zhang et al. (3), and KMERF produces an unbiased kernel correlation, such that chi-square test is a valid test of independence under proper $\alpha$.*
> > >
> > > Therefore, despite being an adaptive kernel construction, the random forest proximity kernel still adheres to the established properties within the existing kernel literature. Therefore, both the permutation test and the chi-square test can be used in KMERF as a valid test. Specifically, when $X$ and $Y$ are independent, the random forest construction effectively results in a random partition of $X$, as $Y$ does not carry any information about $X$, and the resulting kernel correlation converges to $0$ under independence.
> > > Consequently, the kernel correlation derived from the random forest and the resulting chi-square test should not cause inflation in the test statistic nor lead to any issues with post-selection inference.
> > >
> > > **References:**
> > > 1. Phillip Good, "Permutation, Parametric, and Bootstrap Tests of Hypotheses", Springer, 2005.
> > > 2. Cencheng Shen, Sambit Panda, Joshua Vogelstein, "The chi-square test of distance correlation", Journal of Computational and Graphical Statistics, 2022.
> > > 3. Qinyi Zhang, Sarah Filippi, Arthur Gretton, Dino Sejdinovic, "Large-scale kernel methods for independence testing", Statistics
> > > and Computing, 2018.

---

### Official Review · Reviewer_CaNi · 2023-07-05

**Soundness:** 3 good
**Presentation:** 3 good
**Contribution:** 3 good
**Rating:** 6
**Confidence:** 4

**Summary:**

In this paper, random forest induced kernel/proximity is combined with distance correlation and a recently developed chi-square test method to form a hypothesis testing method that is useful for independence testing and k-sample testing. The authors prove that the kernel is asymptotically characteristic and therefore the proposed method is valid and consistent for a sufficiently large sample size. Through experimental evaluation of statistical power for independence and two-sample testing on synthetic data, the authors show that their method performs better than other tests for the majority of simulations settings, and is good at identifying important features. When applied to real biomarker data, the method successfully identifies a potentially valuable marker for pancreatic cancer.

**Strengths:**

- The method is easy to implement and seems powerful. It inherits the advantages of random forest, e.g., working very well on small datasets, almost no need for parameter tuning and preprocessing, easy-to-access high-quality implementations, etc.
- The paper is well-written and easy to follow.
- Theoretical properties of the method are investigated. The results seem correct.
- The chi-square test is much more efficient than the permutation test.
- Theorem 2 seems novel.
- Experimental evaluation is well conducted and looks convincing.

**Weaknesses:**

- The novelty of this method is limited because the combination is straightforward. The random forest induced proximity is well-known in the literature.
- The related work on random forests for hypothesis testing is not surveyed.

**Questions:**

- When does the premise of Theorem 2 (part area goes to zero) hold?
- What are the labels for supervised RF training?

**Limitations:**

The limitations of this work have been discussed in the Appendix.

---

> ### Author Rebuttal · Authors · 2023-08-09
>
> Thank you very much for the review and the questions. Indeed, the main innovation is not about the random forest kernel itself, as the proximity matrix is well-established within the random forest framework. Our main contribution is the direct utilization of this proximity matrix as a valid and consistent kernel choice for hypothesis testing – a approach not explored in existing literature. In the realm of kernel testing, it is widely recognized that different kernels can yield distinct testing performance outcomes. Please find the proposed edits below on how we intend to expand the random forest background in the introduction, as well as to emphasize the main contribution in the conclusion.
>
> **Question 1:**
>
> Thank you for the question! This is indeed an important component that should have been clarified. We will include the following explanation after Theorem 2:
>
> > Note that a crucial assumption in this context is that the area of each leaf region approaches zero. This particular asymptotic behavior is generally met by most continuous and smooth random variables when employing standard random forest. For instance, consider a scenario where the sample data has a continuous distribution within the support. In the default configuration of a random forest, each leaf node accommodates a relatively small number of observations, typically 1 in regression or 5 in classification. Consequently, as the sample size increases to infinite, the area of each leaf node converges to zero.
>
>
> **Question 2:**
>
> The simulations considered two types of hypothesis tests: testing for independence and testing for two-sample. For the RF training on two-sample testing, the labels are binary, specifically 0 and 1. Here, 0 signifies that the samples belong to one group, while 1 indicates that the samples belong to another group. Then, for RF training on independence testing, the labels are simply Y, representing a 1-dimensional response variable throughout the simulations.
>
>
> **Proposed Edits to Introduction and Conclusion:**
>
> Here is the proposed changes for the first paragraph in the introduction section (Line 23-24) to provide a better review of random forest works:
>
> > …This proximity matrix functions as an induced kernel or similarity matrix for the decision forest. The connection between random forest and the induced kernel is well-established, with research demonstrating that random forest can essentially be perceived as a kernel method, generating kernels that surpass conventional ones [1,2, 3]. Generally, any random partition algorithm has the potential to generate a proximity matrix and be conceptualized as a kernel.
>
> Then by the end of the third paragraph (Line 44) we will add the following on existing works regarding random forest for testing.
>
> > ... There are existing studies that utilize random forests for hypothesis testing, such as F-test and feature screening [4], as well as two-sample testing [5]. However, these studies utilize other random forest outputs rather than the proximity matrix. To the best of our knowledge, there has been no research that explores the characteristic properties of the induced kernel, nor any work that directly employs the induced random forest kernel for tasks related to independence or two-sample testing.
>
> Then in the discussion section at the end of the paper (Lines 198-199), we will emphasize the major contribution, and the potential for other random forest algorithms:
>
> > ... The primary contribution of the paper lies in directly leveraging the induced kernel from random forest for achieving consistent and valid hypothesis testing for independence and two-sample. We explored the theoretical properties of this approach while showcasing its numerical benefits. With the aid of the recent chi-square test procedure for kernel-based tests, the complexity of our proposed method is $\mathcal{O}(n^2)$ to compute the test statistic and p-value. Additionally, owing to the data-adaptive nature of the random forest algorithm, the resulting kernel appears to effectively adjust to the inherent data structure, thereby enhancing testing power. Other kernels, whether derived from variations of the standard random forest [1,2] or from the proximity matrix of alternative random partition or forest algorithms [6, 7], may also be used in the framework.
>
> **Additional references:**
> 1. Alex Davies, Zoubin Ghahramani, “The Random Forest Kernel and creating other kernels for big data from random partitions”, arXiv:1402.4293.
> 2. Erwan Scornet, “Random forests and kernel methods”, arXiv:1502.03836.
> 3. Gerard Biau and Erwan Scornet, "A random forest guided tour”, TEST, 2016.
> 4. Tim Coleman, Wei Peng, Lucas Mentch, “Scalable and Efficient Hypothesis Testing with Random Forests ”, Journal of Machine Learning Research.
> 5. Simon Hediger, Loris Michel, Jeffrey Näf, “On the use of random forest for two-sample testing”, Computational Statistics and Data Analysis, 2022.
> 6. Tyler Tomita et al., “Sparse Projection Oblique Randomer Forests”, Journal of Machine Learning Research, 2020.
> 7. Pierre Geurts, Damien Ernst, Louis Wehenkel, “Extremely randomized trees”, Machine Learning, 2006.

---

> > ### Comment · Reviewer_CaNi · 2023-08-20
> >
> > Thanks for your detailed response. I will keep the score unchanged.

---

### Official Review · Reviewer_zSKe · 2023-07-06

**Soundness:** 3 good
**Presentation:** 3 good
**Contribution:** 2 fair
**Rating:** 6
**Confidence:** 4

**Summary:**

In this study, the authors proposed a new kernel KMERF for independence testing.
In KMERF, multiple decision trees are constructed similar to Random Forest, and the number of trees in which two data points belong to the same leaf node is calculated.
This count is used as the kernel value between the two points to form a kernel matrix.
The p-value is then computed using a method similar to HSIC.
The authors proved that KMERF is a characteristic kernel in the limit as the number of data points and the number of decision trees approach infinity.
This property guarantees that KMERF is an effective kernel (asymptotically) for independence testing.

Through experiments using synthetic data, the authors reported that independence testing with KMERF outperforms other kernel-based methods in terms of statistical power.
They also showed that by estimating the feature importance of the Random Forest, they can estimate the input features contributing to the independence testing.
This ability of estimating feature importance is considered an advantage of KMERF.

**Strengths:**

The strengths of this study is on the theoretical anaysis of KMERF and the experimental results.
For valid independence testing, it is required to demonstrate the (asymptotic) characteristic kernel property of KMERF.
The theoretical proof in this study is therefore essential.
Moreover, the experiments report that the testing using KMERF exhibits higher statistical power compared to other methods using different kernels.
Additionally, a unique feature of KMERF is its ability to estimate the important features contributing to the testing by examining the feature importance of the Random Forest.

**Weaknesses:**

A weakness of this study is the absence of a review of existing random forest-based kernels.
For example, [7] presents a kernel based on random partitions similar to KMERF.
Moreover, measuring the similarity of two data points using the number of trees with the same leaf node has been used for several data analysis tasks, and packages such as rfProximity have been publicly released.
When compared to these existing methods, Steps 1 and 2 of KMERF can be considered equivalent to rfProximity.

The absence of the aforementioned review poses a problem in properly evaluating the novelty of this study.
In fact, due to the similarity with rfProximity, the novelty of this study lies not in the random forest-based kernel itself (Steps 1 and 2), but rather in the application to testing in Steps 3--5, as well as demonstrating the asymptotic characteristic kernel property of KMERF.

**Questions:**

* Q1. Please reveiw the existing random forest / partition-based kernels.
* Q2. What is the novelty and the advantage of KMERF compared to the kernels in Q1?

---
I would like to thank the authors for the detailed reply. The proposed updates look reasonable.

**Limitations:**

The authors mentioned some possible future directions that are not addressed in the current study.

---

> ### Author Rebuttal · Authors · 2023-08-09
>
> Thank you very much for reviewing the paper and the valuable suggestions!
>
> **Question 1:**
>
> Thank you for pointing out the limited coverage of existing research on random forests. We will expand the background section to include more works of random forest and existing connections to kernel and hypothesis testing. Please find the proposed edits below on how we intend to expand the background.
>
>
> **Question 2:**
>
> Indeed, the core innovation is not about the random forest kernel itself. After all, the proximity matrix is an established and well-known component within the random forest framework. Instead, what sets our work apart is the direct utilization of this proximity matrix as a valid and consistent kernel choice for hypothesis testing – a novel approach not explored in existing literature. In the realm of kernel testing, it is widely recognized that different kernels can yield distinct testing performance outcomes. Therefore, this paper pioneers the potential of random forest kernels for hypothesis testing, and presents compelling theoretical groundwork and satisfactory numerical outcomes. This work also opens up future possibilities for delving deeper into other random forest or random partition methods within the hypothesis testing framework. Please find the proposed edits below on how we intend to emphasize the innovations.
>
>
> **Proposed Edits to Introduction and Conclusion:**
>
> Here is the proposed changes for the first paragraph in the introduction section (Line 23-24) to provide a better review of random forest works:
>
> > …This proximity matrix functions as an induced kernel or similarity matrix for the decision forest. The connection between random forest and the induced kernel is well-established, with research demonstrating that random forest can essentially be perceived as a kernel method, generating kernels that surpass conventional ones [1,2, 3]. Generally, any random partition algorithm has the potential to generate a proximity matrix and be conceptualized as a kernel.
>
> Then by the end of the third paragraph (Line 44) we will add the following on existing works regarding random forest for testing.
>
> > ... There are existing studies that utilize random forests for hypothesis testing, such as F-test and feature screening [4], as well as two-sample testing [5]. However, these studies utilize other random forest outputs rather than the proximity matrix. To the best of our knowledge, there has been no research that explores the characteristic properties of the induced kernel, nor any work that directly employs the induced random forest kernel for tasks related to independence or two-sample testing.
>
> Then in the discussion section at the end of the paper (Lines 198-199), we will emphasize the major contribution, and the potential for other random forest algorithms:
>
> > ... The primary contribution of the paper lies in directly leveraging the induced kernel from random forest for achieving consistent and valid hypothesis testing for independence and two-sample. We explored the theoretical properties of this approach while showcasing its numerical benefits. With the aid of the recent chi-square test procedure for kernel-based tests, the complexity of our proposed method is $\mathcal{O}(n^2)$ to compute the test statistic and p-value. Additionally, owing to the data-adaptive nature of the random forest algorithm, the resulting kernel appears to effectively adjust to the inherent data structure, thereby enhancing testing power. Other kernels, whether derived from variations of the standard random forest [1,2] or from the proximity matrix of alternative random partition or forest algorithms [6, 7], may also be used in the framework.
>
> **Additional references:**
> 1. Alex Davies, Zoubin Ghahramani, “The Random Forest Kernel and creating other kernels for big data from random partitions”, arXiv:1402.4293.
> 2. Erwan Scornet, “Random forests and kernel methods”, arXiv:1502.03836.
> 3. Gerard Biau and Erwan Scornet, "A random forest guided tour”, TEST, 2016.
> 4. Tim Coleman, Wei Peng, Lucas Mentch, “Scalable and Efficient Hypothesis Testing with Random Forests ”, Journal of Machine Learning Research.
> 5. Simon Hediger, Loris Michel, Jeffrey Näf, “On the use of random forest for two-sample testing”, Computational Statistics and Data Analysis, 2022.
> 6. Tyler Tomita et al., “Sparse Projection Oblique Randomer Forests”, Journal of Machine Learning Research, 2020.
> 7. Pierre Geurts, Damien Ernst, Louis Wehenkel, “Extremely randomized trees”, Machine Learning, 2006.

---

> > ### Comment · Reviewer_zSKe · 2023-08-18
> > **Reply**
> >
> > I would like to thank the authors for the detailed reply.
> > The proposed updates look reasonable.
> > I will keep my score.

---

### Author Rebuttal · Authors · 2023-08-09

Thank you all very much for taking time to thoroughly reviewing our paper and providing valuable feedback. Attached is a one page document containing figures we will refer to in each of our rebuttals.

---

### Decision · Program_Chairs · 2023-09-21

**Decision:**

Reject

**Comment:**

The focus of the submission is the independence testing of two random variables in finite-dimensional Euclidean spaces (R^p and R^q). In order to tackle this problem, the authors propose a random forest based kernel. The kernel is claimed to the asymptotically characteristic (Theorem 2).

Independence testing is key problem in data science; as such the focus of the manuscript is relevant. Unfortunately, the submission suffers from multiple serious issues:
1) The manuscript is invalid mathematically straight from the beginning: for arbitrary kernel the mean embedding does not even exist, it is not clear why the authors are talking about separable metric spaces for the domain of the kernel, ...
2) The main statement is similarly problematic as the notion of "asymptotically characteristic"-ness is undefined, ...

Major revision is needed. The authors are suggested to start with a mathematical problem formulation.